# Enumerating regulatory T cells in cryopreserved umbilical cord blood samples using *FOXP3* methylation specific quantitative PCR

Richard C. Duggleby[1,2]*, Hoi Pat Tsang[3,4], Kathryn Strange[1,2], Alasdair McWhinnie[1], Abigail A. Lamikanra[3,4], David J. Roberts[3,4], Diana Hernandez[1,2], J. Alejandro Madrigal[1,5¤], Robert D. Danby[1,2,6]

1 Anthony Nolan Research Institute, London, United Kingdom, 2 UCL Cancer Institute, University College London, London, United Kingdom, 3 National Health Service Blood and Transplant, Oxford, United Kingdom, 4 Nuffield Division of Clinical Laboratory Sciences, Radcliffe Department of Medicine, University of Oxford, Oxford, United Kingdom, 5 UCL Cancer Institute, Royal Free NHS Trust, London, United Kingdom, 6 Department of Haematology, Oxford University Hospitals NHS Foundation Trust, Oxford, United Kingdom

¤ Current Address: UCL Cancer Institute, Royal Free NHS Trust, London, United Kingdom.
* Richard.Duggleby@anthonynolan.org

**Data Availability Statement:** All relevant data are within the manuscript and its Supporting Information files. Additionally, for Fig 1A and S1

## Abstract

### Background

Allogeneic haematopoietic cell transplantation (HCT) is a curative therapy for severe hae-matological disorders. However, it carries significant risk of morbidity and mortality. To improve patient outcomes, better graft selection strategies are needed, incorporating HLA matching with clinically important graft characteristics. Studies have shown that the cellular content of HCT grafts, specifically higher ratios of T regulatory (Tregs)/T cells, are important factors influencing outcomes when using adult peripheral blood mobilised grafts. So far, no equivalent study exists in umbilical cord blood (CB) transplantation due to the limitations of cryopreserved CB samples.

### Study design and methods

To establish the most robust and efficient way to measure the Treg content of previously cryopreserved CB units, we compared the enumeration of Treg and CD3+ cells using flow cytometry and an epigenetic, DNA-based methodology. The two methods were assessed for their agreement, consistency and susceptibility to error when enumerating Treg and CD3 + cell numbers in both fresh and cryopreserved CB samples.

### Results

Epigenetic enumeration gave consistent and comparable results in both fresh and frozen CB samples. By contrast, assessment of Tregs and CD3+ cells by flow cytometry was only possible in fresh samples due to significant cell death following cryopreservation and thawing.

Fig, the raw data and analysis, along with the machine settings can be found at https://flowrepository.org ID: FR-FCM-Z2DR and FR-FCM-Z2V4.

**Funding:** The Anthony Nolan Research Institute is part of the Anthony Nolan, a registered charity (803716/SC038827). Authors Hoi Pat Tsang, Abigail A. Lamikanra, and David J. Roberts were supported by NHS Blood and Transplant intra mural funding.

**Competing interests:** The authors have declared that no competing interests exist.

## Conclusion

Epigenetic assessment offers significant advantages over flow cytometry for analysing cryopreserved CB; similar cell numbers were observed both in fresh and frozen samples. Furthermore, multiple epigenetic assessments can be performed from DNA extracted from small cryopreserved CB segments; often the only CB sample available for clinical studies.

## Introduction

### Regulatory T cells

Regulatory T cells (Tregs) are a crucial subset of $CD4^+$ T cells that are key regulators of peripheral immune tolerance and essential for maintaining immune homeostasis. As no single definitive cell surface marker for Tregs has yet been identified, most studies use multi-colour flow cytometry to enumerate Tregs, using a combination of cell surface markers (CD4, CD25, CD127) and intracellular FOXP3 [1]. However, this phenotype has some overlap with activated T cells, which may transiently express FOXP3 [2].

### Tregs in haematopoietic cell transplantation

It has become apparent that accurate enumeration of Tregs is particularly important in clinical studies of allogeneic haematopoietic cell transplantation (HCT), where donor Tregs have an important role in immune tolerance between recipient and donor. In 2016, Danby *et al.*, measured Tregs, using multi-colour flow cytometry, in freshly isolated, mobilised peripheral blood stem cell (PBSC) grafts from healthy donors [3]. In this prospective study, higher Treg/T cell (CD3+ cells) ratios in PBSC grafts was strongly associated with improved overall survival and lower non-relapse mortality of patients undergoing allogeneic HCT. Ideally, these studies should be repeated with umbilical cord blood (CB) transplantation to determine if CB Tregs had similar influence. However, in clinical studies of CB transplantation, it is not possible to obtain freshly isolated samples for flow cytometry since CB units are cryopreserved shortly after collection, and it would be impractical to collect samples of the thawed CB unit at the time of infusion. Therefore, the only samples available for assessment of cryopreserved clinical grade CB units are small segments (blood bag tubing containing less than 200 µl of blood), used for quality control purposes [4]. Previous studies have shown that the cell content of these segments, when analysed by flow cytometry [5], show significant variance from the parent unit post thaw, as cell viability in the segments and units is very variable [6].

### Using epigenetic signatures to measure Treg numbers

*FOXP3* gene expression is regulated through epigenetic modification by methylation of DNA at cytosine-guanine (CpG) motifs [7]. Methylation of CpG indicates chromatin condensation, whilst demethylation leads to relaxation of the chromatin and greater accessibility of the target region to the transcription machinery i.e. expression of methylated genes is repressed [7]. The methylation status of CpG dinucleotides can be assessed by bisulphite conversion; a process whereby demethylated cytosines are converted to uracil, whilst methylated CpG bases remain protected. Quantitative polymerase chain reaction (qPCR) primers can be designed that bind to either the bisulphite-specific methylated or demethylated sequence (methylation specific), allowing for detection and quantification within a specific genomic DNA locus [8].

Using such methods, Baron *et al.*, analysed the DNA methylation status of CpG dinucleotides within *FOXP3* [9]. In natural Tregs, at a specific position within the first intron of *FOXP3* (Treg specific demethylated region, TSDR), the CpG dinucleotides were demethylated, allowing for stable *FOXP3* gene expression, but remained methylated in all other major peripheral blood cells, including activated T cells [9]. Hence, the TSDR methylation status can distinguish natural/thymic Tregs from other cells. Similarly, methylation specific regions have since been identified for other cell types and genes regulated by methylation, such as *CD3* (T cell specific demethylated region, TcSDR) [10]. The advantage of such techniques is that they allow epigenetic enumeration of cells in samples that have been stored in conditions unfavourable for live cell analysis, such as solid tissue [10] or cryopreserved samples [11].

In this study, we validated the use of a DNA methylation status-based method to measure the Treg/CD3+ cell ratio in cryopreserved samples of CB or archived DNA from CB units, which are readily available for retrospective studies or for testing ahead of CB transplantation without compromising the clinical graft. In order to validate the methodology, we first compared the results of Treg enumeration in freshly obtained CBUs using standard flow cytometry to that of the methylation status-based technique. Once satisfied that the two methods were comparable with fresh samples, we proceeded to enumerate Treg/T cell ratios in cryopreserved segments and archived DNA which are not amenable to flow cytometric analysis. We believe that this method would be a fast and efficient way of carrying out a retrospective study to assess the impact of Treg/T cell ratio on patient outcomes after HCT. If a strong clinical relationship is present, it may allow the use of Treg/T cell ratio as an additional criterion for optimal cord blood unit selection.

## Materials and methods

### Study population

Between March 2018 and Feb 2019, fresh umbilical CB units, within 48 hours of collection, and frozen CB segments were supplied by the Anthony Nolan Cell Therapy Centre, Nottingham. Informed written consent from the mothers had been obtained and ethical permission for collection. Approval was obtained from the East Midlands Derby Ethical Committee (15/EM/0045) for the use of non-clinical grade CB units and CB segments for research use. Information on CB unit donors were anonymised as supplied to the Anthony Nolan Research Institute.

Frozen CB segments were thawed using a modification of the method described in Rodriguez *et al.* [12], Segments were thawed and mixed with thawing solution (4˚C, 5% Dextran-40 (Sigma-Aldrich, Dorset, UK), 2.5% AB serum (Lonza, Basel, Switzerland) in phosphate buffered saline (PBS; Lonza)) resulting in a final 1 in 3 dilution of the sample. One third of the final volume (~100 μl) was used for flow cytometry, whilst DNA was extracted from the remaining two thirds (~200 μl).

### Flow cytometry

Both the fresh CB and thawed CB segments were prepared by first lysing the red blood cells (RBC) with BD Pharm Lyse (BD Pharmingen, Berkshire, UK) before surface antibody staining at 4˚C with CD4-FITC (ImmunoTools, Friesoythe; Germany; MEM-241), CD127-PE (BD Pharmingen; HIL-7R-M21), CD25-APC (BD Biosciences; Berkshire, UK; 2A3), CD3-PE-Cy7 (BD Pharmingen; SK7), and CD45-Alexa Fluor 700 (BD Pharmingen; HI30). The cells were washed with serum free PBS and stained with the fixable viability dye eFluro 506 (eBioscience, ThermoFisher, Leicestershire, UK) for 30 mins at 4˚C. The cells were fixed and permeabilised using eBioscience Fixation/Permeabilization solution (eBioscience) and stained with

FOXP3-PerCP-Cy5.5 (eBioscience; PCH101) in accordance with the manufacturer's recommendations. The stained cells were processed on a four laser BD Fortessa (BD Biosciences) and analysed using BD FACS Diva and FlowJo flow cytometry software.

Flow cytometry gating to identify and enumerate T cells and Tregs in CB samples is shown in Fig 1A and S1 Fig, based upon our previously published method [1,13,14]. Leukocytes were gated using side scatter (SSC)/CD45 expression and live cells identified using the fixable viability dye eFluor 506 (Fig 1Ai and 1Aii). Live T cells (SSC$^{low}$/CD3+ cells) were gated (Fig 1Aiii) and Tregs identified based upon CD25, CD127 and FOXP3 expression (Fig 1Aiv–1Aviii). Tregs were defined as CD3$^+$CD4$^+$CD127$^{low}$CD25$^{hi}$FOXP3$^{hi}$ cells [13]. In CB, this population has a relatively low expression of FOXP3 (Fig 1Avii); CB Tregs are resting and naïve [15–17]. To allow accurate identification of the FOXP3$^+$ Treg population, CD3$^+$CD4$^+$CD25$^-$ cells were excluded from analysis (Fig 1Avi). The CD127$^{low}$FOXP3$^{hi}$ Tregs were then identified by internal comparison with the CD127$^{hi}$FOXP3$^{int}$ conventional T cells (Fig 1Aviii) [13].

### Methylation specific-quantitative PCR for *FOXP3* and *CD3*

Genomic DNA was extracted from 200 μl of fresh whole CB or ~200 μl of thawed CB using a QIAmp DNA blood mini Kit (Qiagen, Manchester, UK). For frozen samples, the lysis time was extended to 30 mins to improve DNA recovery.

FOXP3+ and CD3+ cell enumeration using TSDR or TcSDR specific qPCR was performed as described by Sehouli *et al.* [10]. In brief, 1 μg of genomic DNA was bisulphite converted using the EZ DNA methylation-Gold Kit (Zymo research, Cambridge Bioscience, Cambridge, UK). Sanger sequencing was performed on selected samples (n = 4) to confirm high efficiency of bisulphite conversion (3730XL sequencer and BigDye reagents; AB applied biosystems, ThermoFisher, Leicestershire, UK). Real time quantitative PCR for measurement of the *FOXP3* TSDR and *CD3* TcSDR was performed on a TaqMan Viia 7 384-well block real-time PCR system (ThermoFisher, Leicestershire, UK). 4 pmols of forward and reverse primers and 4 pmols of MGB TaqMan Probes (AB applied biosystems) per reaction were combined with EpiTect MethyLight PCR master mix (Qiagen), ROX dye (Qiagen) and a minimum of 10 ng of bisulphite converted DNA/well/reaction. Each sample was analysed in quadruplicates for each separate reaction and compared to standard curve of 3–30,000 copies, run in sextuplets (standards provided by NHS Blood and Transplant, Oxford; pMA plasmid constructs containing either CpG sequences for both *FOXP3* and *CD3* targets or TpG sequences of the bisulphite sensitive regions of these genes). Analysis was with Thermofisher applied biosystems tools https://apps.thermofisher.com/apps/spa/.

### Compensating for additional *FOXP3* CpG copies in female samples

In mice and humans, *FOXP3* is located on the X chromosome. In male (XY) samples, Tregs contain a single demethylated (TpG) TSDR, whilst non-Treg cells contain a single methylated (CpG) TSDR. In female (XX) samples, Tregs contain one demethylated (TpG) copy (from the active X-chromosome) and one methylated (CpG) copy (from the inactive X-chromosome). In non-Treg cells, two methylated (CpG) copies will be present. When calculating the proportion of Tregs to non-Treg cells in female samples, a correction factor of 2 was applied to account for the extra methylated (CpG) copies i.e. proportion of Treg *FOXP3* copies = (2$^*$TpG copies)/(TpG copies + CpG copies)).

### Statistical analysis

Statistical analysis was performed on Graphpad Prism v.7 (Prism, GraphPad Software, La Jolla, CA) using the tests stated in the text. Tests for normality are inaccurate on sample sizes

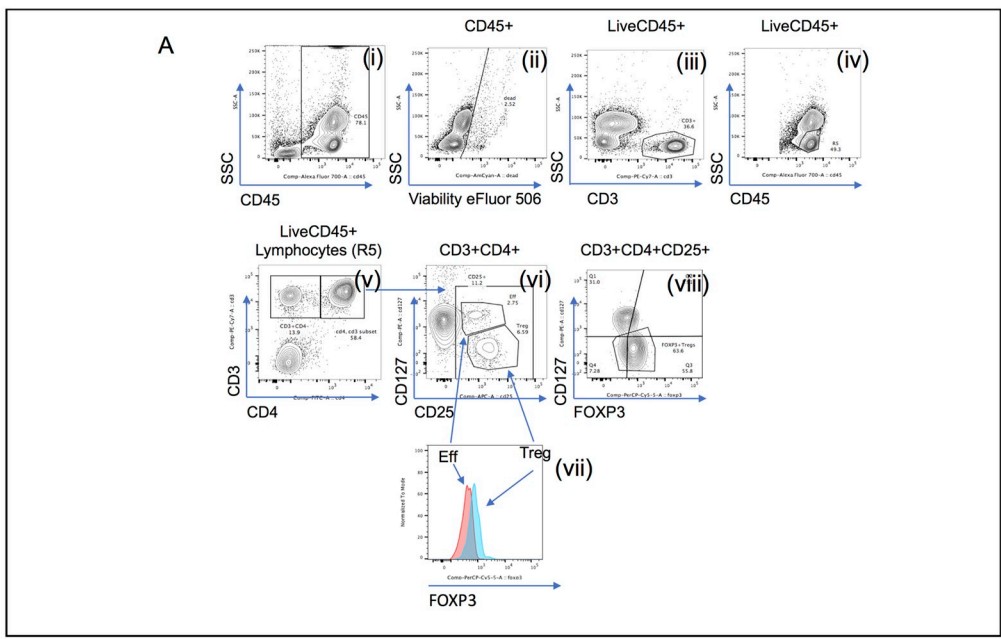

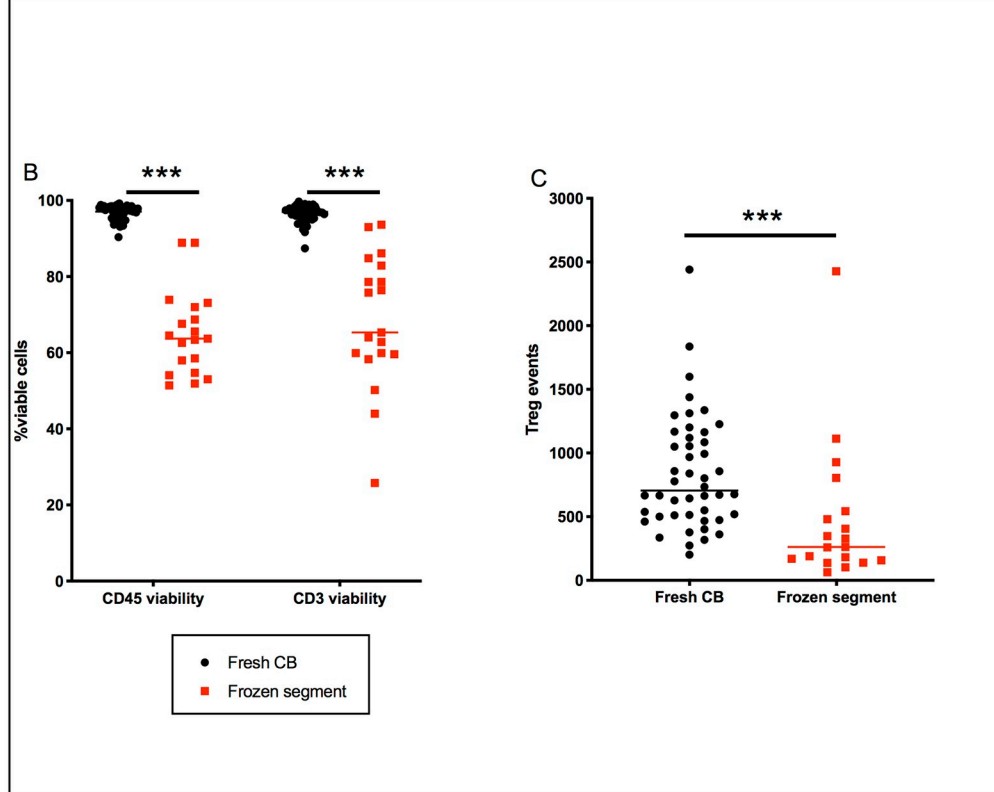

**Fig 1. Flow cytometry gating of CD3+ cells and Tregs with viability and total Treg events. A;** example gating strategy for CD3+, CD3+CD4+ cell and Tregs (CD127$^{low}$FOXP3$^{hi}$) content in fresh CB and segments. Shown is an example from a fresh CB. Fresh CB and thawed segments were RBC lysed and stained for the surface markers CD45, CD3, CD4, CD127, and CD25, followed by the fixable live/dead stain eFluor 506. After washing the cells were fix/permeabilised and stained with FOXP3. Left to right top row; (i) gating CD45+ cells, (ii) excluding dead cells (eFluor506+), (iii) gating SSC$^{low}$/CD3+ cells and (iv) CD45$^{hi}$SSC$^{low}$ lymphocytes (R5). Second row, (v) CD3+CD4+ cells are gated from R5 cells and then (vi) effectors (CD127$^{hi}$) and Tregs (CD127$^{low}$) CD25+ cells. (vii) FOXP3 expression on gated effectors (eff) and Tregs. (viii) gated Tregs from CD127$^{low}$FOXP3$^{hi}$ CD25+ cells. **B;** Proportion of eFluro506$^-$ cells in the CD45$^+$ and CD3$^+$ gated populations fresh CB or frozen segments. **C;** number of live Tregs (as gated 1A) events acquired from fresh CB or frozen segments. Results shown are of non-parametric unpaired Mann-Whitney U tests *** = p ≤ 0.001, ** = p ≤ 0.01 and * = p ≤ 0.05.

such as with the CB segments [18]. Consequently, non-parametric Mann-Whitney U tests were selected with unpaired data and Wilcoxon tests for paired observations, as these do not make an assumption of gaussian distribution.

## Results

### Enumeration of Tregs by flow cytometry

Forty-six fresh CB samples were analysed by flow cytometry. CD45+ cells were 97% viable (median; range, 90–99) and CD3+ cells 97% viable (range, 87–100) (Fig 1B). Expressed as a proportion of CD45+ cells, the median CD3+ and Treg content was 18.6% (range, 8.0–42.0) and 0.82% (range, 0.24–2.84), respectively (Table 1). The Treg/CD3+ cell ratio ranged from 0.02 to 0.1 (median, 0.04).

We then tried to apply the same flow cytometry method to frozen/thawed CB segments but this showed that they had significantly lower cell viability, compared to fresh samples, with only 64% (range, 51–89; p<0.0001) of CD45+ cells being viable and 65% (range, 26–94; p<0.0001) of CD3+ cells (Fig 1B). Whilst the total number of cellular events acquired were similar, cell death resulted in significantly fewer live Treg events in the frozen samples compared to fresh (262 (range, 64–2427) vs 706 (range, 202–2441); p <0.0001, Fig 1C), making reliable Treg enumeration difficult. Furthermore, poor cell viability meant that 12% (3/22) of the frozen/thawed CB samples had to be excluded from analysis due to insufficient Treg events (<100) for reliable flow cytometry quantification.

Discrepancies between the proportion of cells (%CD45+ cells) in frozen vs fresh CB samples were also seen when using flow cytometry, most likely explained by a disproportionate loss of non-T cells in the frozen/thawed segments (CD3+ cells 36.4% vs 18.6%, p<0.0001; Tregs 1.8% vs 0.8%; p<0.0001) (Table 1, Fig 2A and 2B). There was also a small, but significant, difference in the Tregs/CD3+ cell ratio of 0.06 (frozen) vs 0.04 (fresh); p = 0.009 (Table 1, Fig 2C).

We concluded that using flow cytometry for the enumeration of Tregs in cryopreserved CB segments was likely to be unreliable. Thus, we sought to validate the DNA methylation status method instead, as this method does not depend on cell viability.

### Enumeration of T cells and Tregs by DNA methylation analysis

The results of the Treg and T cell (CD3+) enumerations in fresh and frozen CB units by MS-qPCR are shown in Table 1 and Fig 2A–2C. In contrast to flow cytometry assessment, there was no significant difference observed in the proportion of either CD3+ cells (23.2% vs 22.4%, p = 0.92), Tregs (1.65% vs 1.40%, p = 0.59) or Treg/CD3+ cell ratio (0.07 vs 0.06, p = 0.6) between fresh and frozen CB samples, when using this method.

**Table 1. Cellular content of fresh CB and frozen segments using flow cytometry or MS-qPCR enumeration.**

|  |  | Fresh CB | Frozen segments |
|---|---|---|---|
| n |  | 46 | 19 |
| % cells (flow cytometry) | CD3+ | 18.57% (8.01–41.98) | 36.44% (7.78–65.45) |
|  | Tregs | 0.82% (0.24–2.84) | 1.76% (0.61–3.66) |
| Ratio (flow cytometry) | Treg/CD3+ | 0.04 (0.02–0.1) | 0.06 (0.02–0.12) |
| % cells (MS-qPCR) | CD3+ | 23.20% (11.72–45.31) | 22.43% (16.30–47.05) |
|  | Tregs | 1.65% (0.61–4.07) | 1.40% (0.69–4.67) |
| Ratio (MS-qPCR) | Treg/CD3+ | 0.07 (0.04–0.19) | 0.06 (0.04–0.21) |

Shown are median values with ranges.

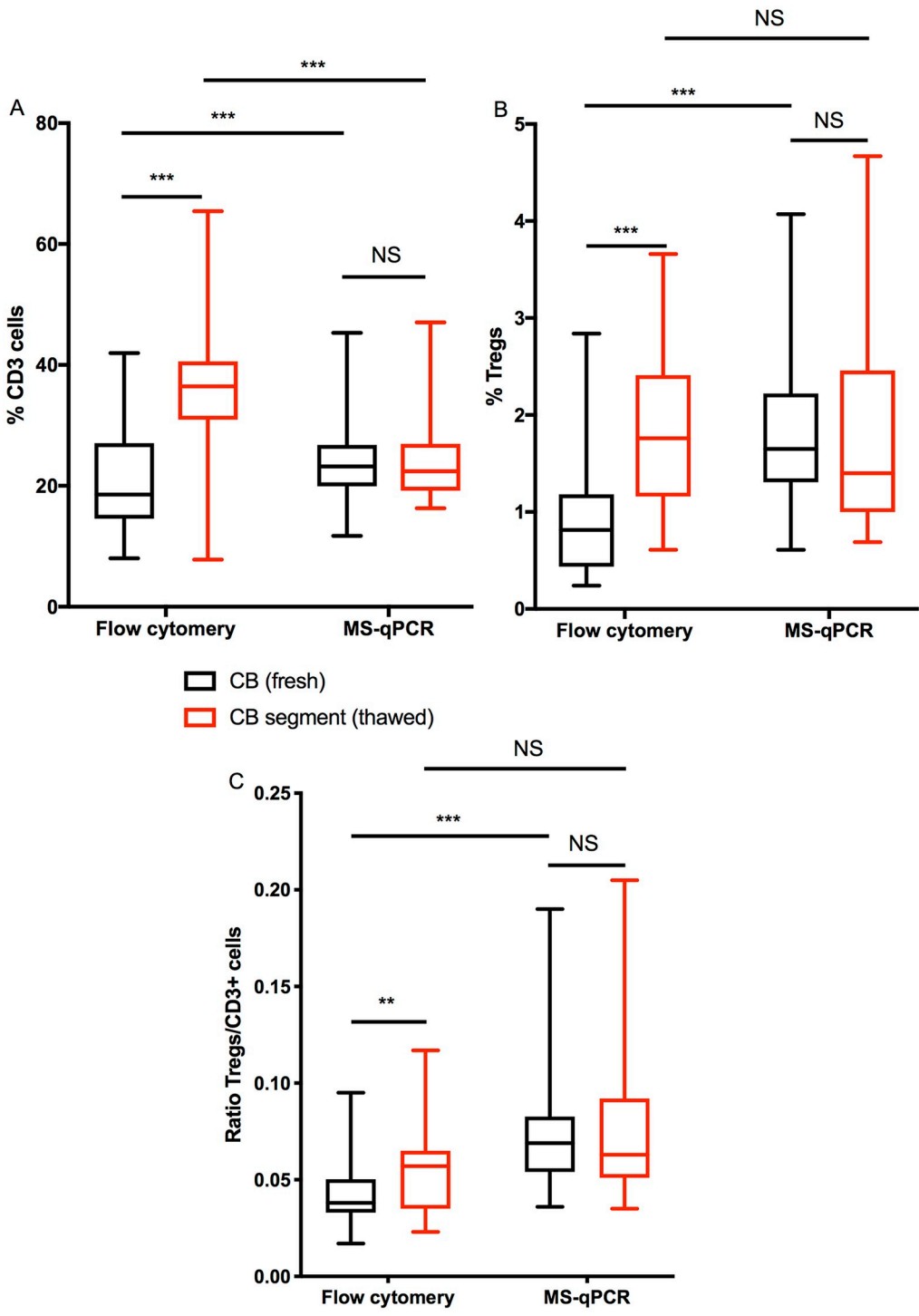

**Fig 2. CD3+ cell and Treg content in fresh CB and segments by flow cytometry and MS-qPCR.** Fresh CB and thawed segments were RBC-lysed and stained for the surface markers CD45, CD3, CD4, CD127, and CD25, followed by the fixable live/dead stain eFluor 506. After washing, the cells were fix/permeabilised and stained with FOXP3. DNA was extracted from paired samples. **A**; proportion of CD3+ cells by flow cytometry (live CD3+ of live CD45+ cells) and by MS-qPCR. **B**; proportion of Tregs by flow cytometry (live Treg of live CD45+ cells) and by MS-qPCR. **C**; Ratio of Treg to CD3+ by flow cytometry and by MS-qPCR. Results of non-parametric unpaired Mann-Whitney U tests or Wilcoxon tests for paired observations *** = p ≤ 0.001, ** = p ≤ 0.01 and * = p ≤ 0.05.

As expected, there was a difference in cell enumeration for any given type of sample depending on the method used; Indicating that the results from either method cannot be combined. In fresh CB samples, the proportions of CD3+ and Tregs measured by MS-qPCR were higher than those measured by flow cytometry CD3+ (23.2% vs 18.6%, p = 0.0006) Tregs (1.65% vs 0.82%, p<0.0001), whilst in frozen CB samples, the proportions of CD3+ cells were lower (22.4% vs 36.4%, p = 0.0004) but the Tregs remained the same. (Table 1 and Fig 2). Importantly we were able to confirm that the proportions of T-regs and the ratios of T-regs to T-cells in different cord units were variable.; the Treg to CD3 ratio in fresh samples by flow cytometry was as high 0.1 and as low as 0.02, whilst by MS-qPCR, it was as high as 0.19 and as low as 0.04. Indicating that it was possible to observe a 5 fold difference in the Treg to CD3 ratio between different individuals.

## Assessment of T cells and Tregs in male and female CB samples

As *FOXP3* is located on the X chromosome, a correction factor needs to be applied to female samples as they have two copies of the gene. However, Treg enumeration results by MS-qPCR analysed separately for males and females indicated differences, as in other published studies [19,20], even with the correction factor. First, we discounted any biological differences between the genders. To do this, we compared the proportion of CD3+ cells (%total cells), using either MS-qPCR or flow cytometry (Fig 3Ai and 3Aii) or the proportion of Tregs (%total cells) using flow cytometry (Fig 3Aiii) (which are not influenced by X-linkage) between the genders and found no significant differences between them in fresh samples. We then analysed the MS-qPCR Treg enumeration data after applying the correction factor to female samples, and found (Fig 3Aiv) that the Treg proportions (%total cells) were significantly higher in female samples compared to male samples both in fresh (2.11% (1.21–4.07) vs 1.43% (0.61–2.76), p = 0.0004) and frozen samples (Fig 3B), (1.85% (1.0–4.7) vs 1.13% (0.7–2.5), p = 0.049) (Fig 3Biv). This is likely due to the correction factor assuming 100% X-inactivation in individual female cells, which maybe incomplete.

## Correlation between flow cytometry and MS-qPCR analysis

Despite the proportions of Tregs and T cells in fresh samples being different, depending on the method used, there is a clear relationship between the two measurements. In paired analysis of 46 fresh CB samples (24 female, 22 male), there was a strong correlation when enumerating T cells (as %total cells) by flow cytometry and the MS-qPCR assay (r = 0.63, p<0.0001) (Fig 4A). This correlation was maintained in subgroup analysis of the male and female samples separately (r = 0.63, p = 0.002 and r = 0.64, p = 0.0008, respectively). Similarly, a significant correlation was observed when Tregs (%total cells) were enumerated using the two methods (r = 0.32, p = 0.03) (Fig 4B). In contrast to the measurement of CD3+ cells, the significant correlation present in the male samples (r = 0.57, p = 0.006), was absent in the female samples (r = 0.19. p = 0.39). When analysing the Treg/CD3+ cell ratio, results from the flow cytometry and MS-qPCR assays showed a significant correlation for the male samples only (r = 0.54, p = 0.009). A correlation was not observed for either the combined samples (r = 0.12, p = 0.44) or female samples alone (r = -0.13, p = 0.53). Of note, the ratio of Tregs to T cells was consistently higher when measured by MS-qPCR compared to flow cytometry across all samples.

Overall, these findings demonstrate that there is a strong correlation when enumerating CD3+ cells between the two techniques, when enumerating Tregs or the Treg/CD3 ratio in fresh CB samples from male donors. By contrast, the variability in the flow cytometry assessment of Tregs meant that there was a poor relationship between the two methods with 19

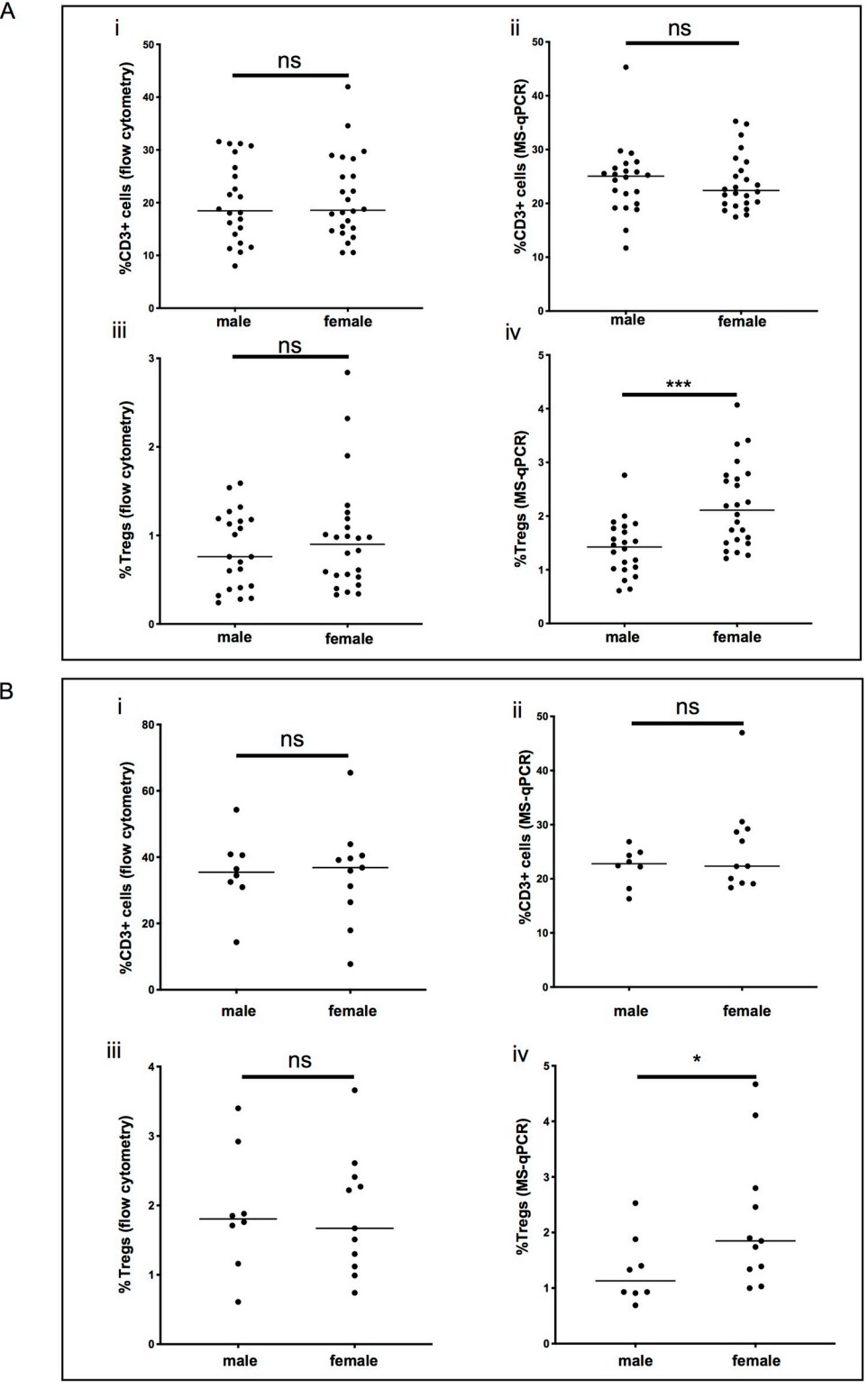

**Fig 3. Comparison of CD3+ cell and Treg enumeration in male and female samples. A;** Comparison between male and female fresh samples. **B;** comparison between male and female samples from frozen segments. (**i**); Proportion of CD3+ cells measured by flow cytometry (**ii**); Proportion of CD3+ cells by TcSDR MS-qPCR enumeration. (**iii**); Proportion of FOXP3+ cells by flow cytometry and (**iv**); by TSDR MS-qPCR enumeration. Shown is the results of Mann-Whitney U tests *** = p ≤ 0.001, * = p ≤ 0.05.

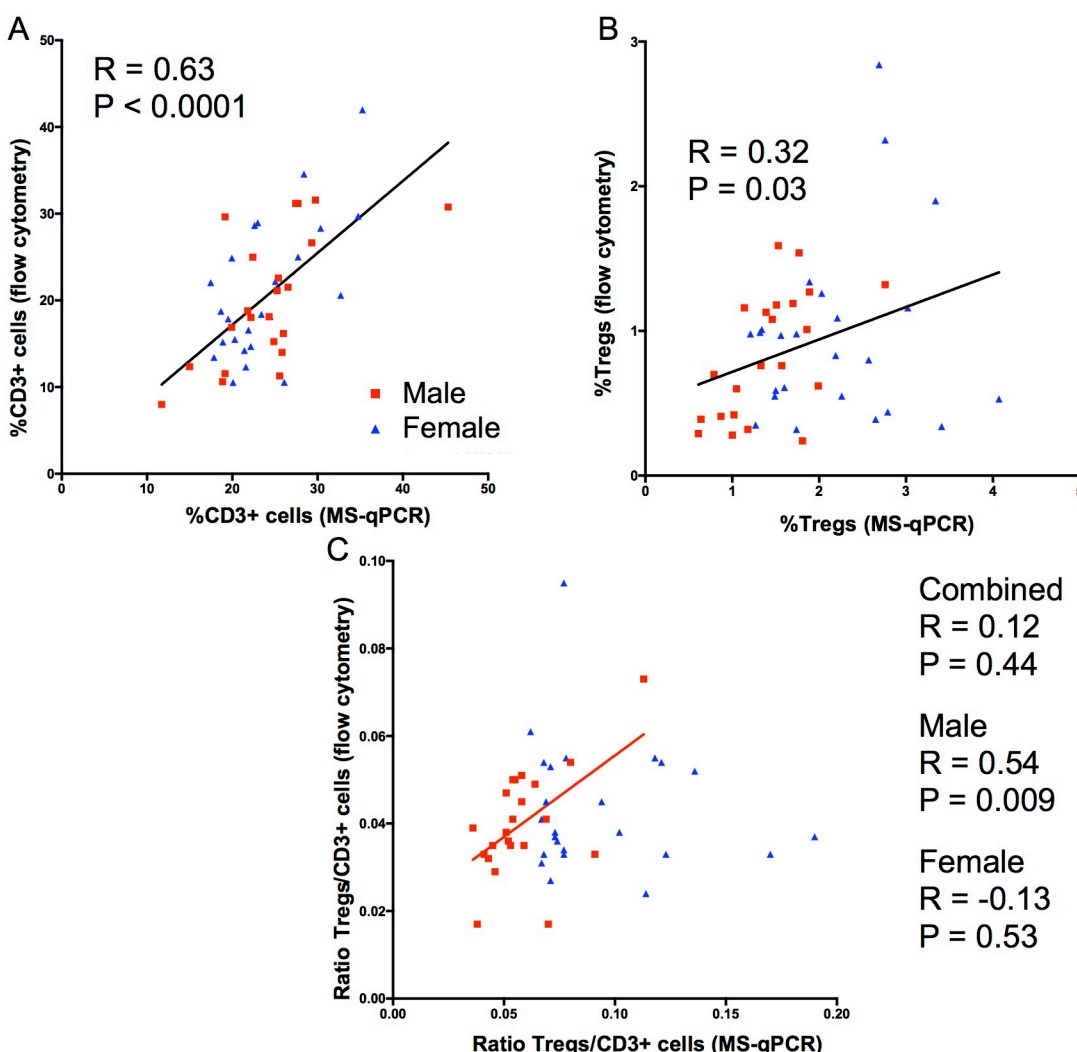

**Fig 4. Comparison of flow cytometric and MS-qPCR data for CD3+ cell, Treg content, and Treg/CD3+ cell ratio in fresh samples.** Cellular enumerations, from fresh CB samples, using flow cytometry were compared with epigenetic enumerations performed on the same samples. Male and female derived samples are as indicated. The results of a Pearson's correlation coefficient are shown. **A;** Proportion of CD3+ cells by flow cytometric or with TcSDR MS-qPCR enumeration. **B;** Proportion of Treg by flow cytometry (as gated in Fig 1A) or with TSDR MS-qPCR enumeration **C;** Ratio of Treg/CD3+ cells using the two methods.

frozen CB samples (eight males, eleven females); the proportion of Tregs and the Treg/CD3+ cell ratio showed no correlation between the two methods in frozen samples (Fig 5).

## Discussion

The purpose of this study was to evaluate alternative methodologies for the enumeration of Tregs in cryopreserved cord blood samples, to allow retrospective studies on the impact of CB Tregs on the outcomes of HCT. Traditionally flow cytometry is used to enumerate different cell populations in blood samples. However, in practice, this method is best suited to high quality, fresh cell samples to provide accurate and reliable results. Obtaining such quality from cryopreserved, archived cord blood segment samples is challenging and, therefore, alternative methodologies could provide a way to carry out such studies.

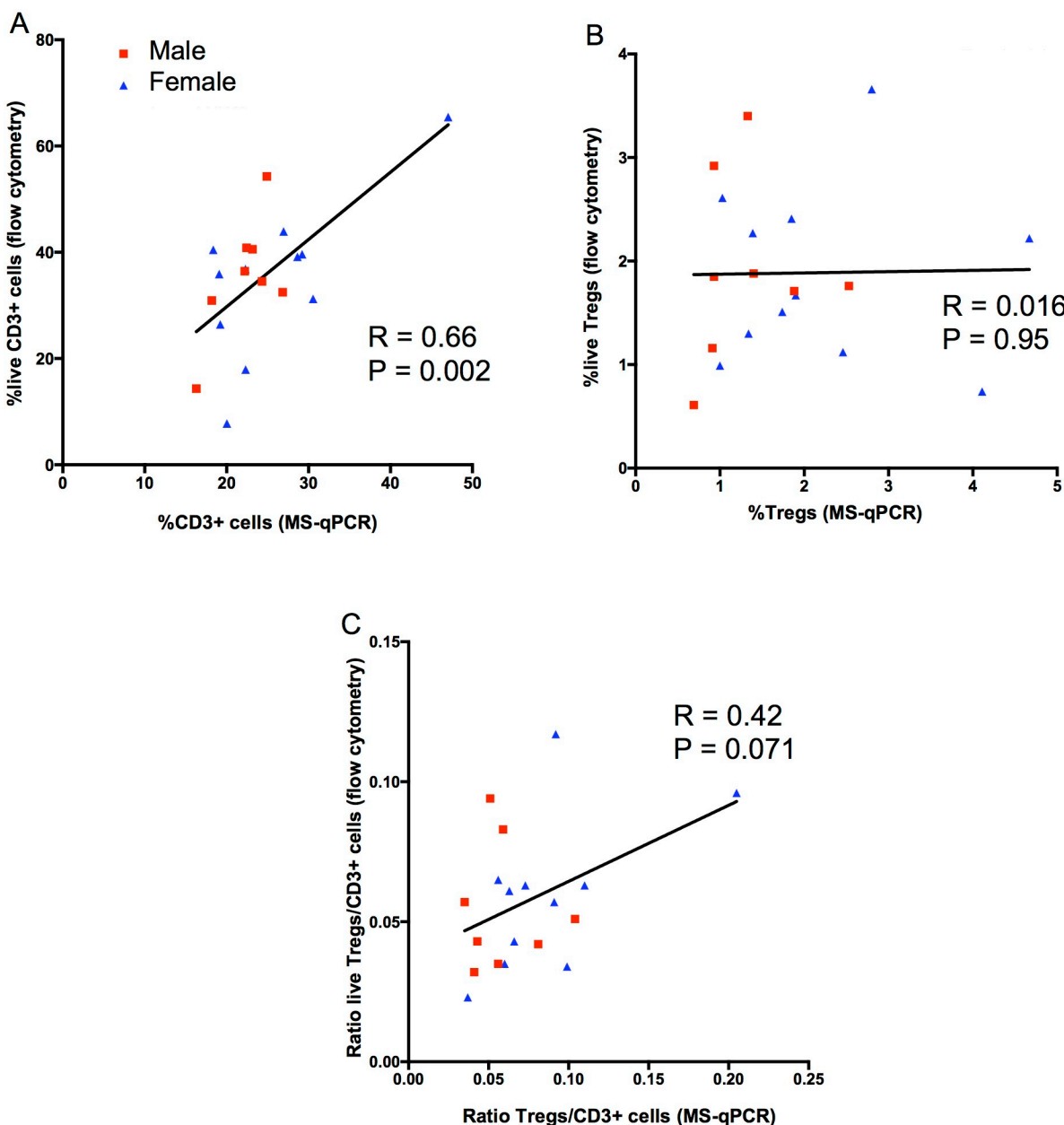

**Fig 5. Comparison of flow cytometric and MS-qPCR data for CD3+ cell, Treg content, and Treg/CD3+ cell ratio in frozen samples.**
Cellular enumerations, from frozen CB segments, using flow cytometry were compared with epigenetic enumerations performed on the
same samples. Male and female derived samples are as indicated. The results of a Pearson's correlation coefficient are shown. **A;** CD3+ cells
by flow cytometric or with TcSDR MS-qPCR enumeration. **B;** Treg by flow cytometry or with TSDR MS-qPCR enumeration. **C;** Ratio of
Treg/CD3+ cells using the two methods.

The first aim of this study was to validate the use of the MS-qPCR DNA-based method to
assess the Treg/CD3 ratios in fresh CB samples, as has been performed in adult peripheral
blood [11]. In fresh samples, where both flow cytometry and the DNA MS-qPCR methods
could be used reliably, we found that although the proportions of Treg and CD3 of total cells
obtained by each method were different, the differences were mostly consistent (the numbers
were always higher when measured by MS-qPCR); i.e. the measures were still related. These

differences can be explained by the nature of the methodologies. Flow cytometry gating strategies tend to progressively narrow down the population of interest from a pool by excluding ambiguous events from the counts. This can, as a consequence, lead to underestimation of the true counts. DNA based methods on the other hand do not exclude cells (for example dead cells).

Of note, the correlation between Treg numbers assessed by flow cytometry and by the MS-qPCR assay in fresh samples is similar to that previously reported by Liu *et al.* [21]. Our ultimate aim is to compare the Treg to CD3 ratios between CB samples from different donors. Thus, as long as the same method is employed to measure all the samples, either technique could be used in samples when viability is not compromised. By contrast, the MS-qPCR method could be used in samples where flow cytometry is not possible. Regardless of the methodology used, we observed sufficient variation between samples to justify studying the correlation between Treg/T cell ratios and HCT outcomes.

A strong correlation was found between the CD3+ cells quantified by MS-qPCR and by flow cytometry in fresh samples. However, with the Treg/CD3+ cell ratio, only the male samples showed a correlation between the two methods. Additionally, there was a clear increase of Tregs with the MS-qPCR assessment in female samples, and greater noise in the assessment from female DNA samples. This bias is not present in the flow cytometry assessments and also not in MS-qPCR CD3+ cell enumeration. This suggests a sample bias in the estimation of *FOXP3* copy number by MS-qPCR assay alone in female samples. In the assay, in female samples, we correct for copies of inactive, methylated *FOXP3* in the Treg cells as these are from the *FOXP3* gene present on the inactive X-chromosome, made chromatically silent by X-chromosome inactivation (XCI). However, using this correction there seemed to be significantly a higher proportion of demethylated *FOXP3* in the female samples compared with the male samples. This is not unprecedented, as a number of other studies have observed similar effects in other tissues [19,20]. In these studies, *FOXP3* was the only epigenetic measure, but in our study, *CD3* enumeration is also used. Since the *CD3* gene is not on the X-chromosome and, therefore, not subjected to XCI, this indicates that the shift in the female demethylated *FOXP3* enumeration is not due to experimental error, but rather something inherent to *FOXP3* in female samples. One possible explanation is that the XCI in the Tregs is not 100%, leading to less methylated *FOXP3* and more demethylated TSDR *FOXP3*.

Studies of XCI indicate that as many as 15% of genes escape XCI, to varying degrees, and are expressed from the inactivate X-chromosome. This results in higher copy numbers in female samples [22]. Variabilities in the levels of XCI have been found in both myeloid and lymphoid lineages and particularly in naïve lymphocytes [23,24]. With Tregs, XCI acts as a form of quality control of the *FOXP3* gene; for example in the rare disease linked to the dysfunction of FOXP3, immunodysregulation polyendocrinopathy enteropathy X-linked (or IPEX) syndrome, IPEX females are heterozygous for the mutant *FOXP3* and it is silent [25]. Taken together this suggests that the shift in demethylated *FOXP3* in female samples is likely due to variable levels of XCI.

The secondary aim was to determine if a DNA based method was practical with the frozen CB segments. It was clear that whilst the enumeration of Tregs in fresh CB units with flow cytometry was robust, there were clear difficulties with frozen samples. This resulted in skewed populations post-thaw that were not observed using MS-qPCR. This ultimately resulted in a poor agreement between the two methods with thawed samples.

One of the main considerations when enumerating T cells and Tregs in frozen CB units by flow cytometry is limited starting material. Assessments of the CB units in storage are limited to the 100–150 µl segments. This is coupled with high cell death during freezing and thawing, as demonstrated in previous studies [6,26,27].

On comparing the flow cytometry data from thawed CB to that of the fresh CB units, extensive cell death was present in the thawed segments. As a result, flow cytometry assessment of thawed segments detected significantly less live gated Treg events when compared to fresh samples; a potential negative impact on the accuracy of flow cytometry and requiring more starting material. With the DNA based MS-qPCR assay, multiple tests could potentially be performed on a single sample; the median amount of DNA extracted from a thawed CB segment was 7.3µg (range 2.9–13.3; S2 Fig) and each MS-qPCR assay required 1–2µg of genomic DNA.

As the MS-qPCR method enumerates the proportion of total CD3+ cells or Tregs, live or dead, this does raise the question of whether gating for total cells by flow cytometry overcomes these problems. Thus, for completeness, the live gate was omitted before gating for the CD3+ and Treg populations (S1 and S3 Figs). However, whilst the correlation between the CD3+ cell enumeration methods showed reduced scatter (S3A Fig), total cell gating of the flow cytometry assessments did not improve the correlation with the Treg data (S3B and S3C Fig). Thus, the conclusion remains the same.

Previously published studies have also described how cell death in frozen CB has caused practical issues for flow cytometry [6,26,27]. In our study, there is clear evidence that the cellular population has become skewed post thaw. As granulocytes are particularly sensitive to loss of viability [28] the differences in fresh and frozen cell proportions can, at least in part, be explained by loss of CD3 negative cells. However, this does not account for differences in the Treg/CD3+ cell ratios.

Other studies have examined the effect of cryopreservation on Tregs enumeration [29–31] and there is evidence that cryopreservation negatively impacts on the ability to accurately gate Tregs by flow cytometry [31]. It should be noted, however, that the effect observed here is small, especially compared to relative increases in the Tregs and CD3+ cells of total live CD45+ cells. Therefore, it is possible that this is an artefact of reduced Treg numbers in the live gate and the relatively limited number of CB samples analysed in our study. Ideally, flow cytometry analysis of paired observations from CB units before and after cryopreservation should be performed to see whether this effect is still maintained. Only a limited number (n = 10) of paired DNA samples from frozen segments and their parent unit (before processing) were available for FOXP3 and CD3 assessment using MS-qPCR (S4 Fig). However, no significant difference was observed between the pairs for Tregs, CD3 or ratio (S4 Fig), replicating the observation made with unpaired DNA samples.

Having established that the MS-qPCR based enumeration is a good alternative to flow cytometry-based enumeration in fresh samples (especially in male samples), the same analysis was then applied to frozen samples. The MS-qPCR enumeration showed little difference when analysing fresh and frozen samples as it is not dependent on cell viability, unlike flow cytometry. This was reflected in the similar numbers of Tregs and CD3+ cells observed in fresh and frozen samples.

In conclusion, data from the assessment of fresh CB samples, where event numbers and viability are not limiting factors, show a strong relationship between the flow cytometry and MS-qPCR assessment. For frozen CB segments, flow cytometry enumeration is less reliable due to high cell death and limited starting material. By contrast, the MS-qPCR assessment appears unaffected by cell death or the limited size of the segments. Our study did highlight, however, that incomplete XCI means that Treg enumerations from male and female samples should be treated separately. Ultimately, we have demonstrated that it is possible to use a DNA based method to assess the Treg/CD3+ cell ratio, even with samples stored in suboptimal conditions where flow cytometry is not reliable. This technique will also allow for cell enumeration in historical samples where only DNA is available. A retrospective study, similar to that performed on fresh samples with adult PBSCs, is therefore possible by using the MS-qPCR assays [3]. The

DNA based method, whilst being robust, does not account for cell death in clinically used samples. The question then would be if the influence of the Treg/CD3+ cell content in these transplanted units is greater than the potential experimental noise introduced? If so, the Treg/CD3+ cell ratio could become a novel selection criterion for banking and selecting CB units for clinical transplantation.

## Supporting information

**S1 Fig. Example flow cytometry of a thawed CB segment with and without exclusion of dead cells.** The same gating strategy as applied in Fig 1 but with an example thawed CB segment. Left to right top row; (i) gating CD45+ cells, (ii) excluding dead cells (eFluor506+), (iii) gating SSC$^{low}$/CD3+ cells and (iv) CD45$^{hi}$SSC$^{low}$ lymphocytes (R5). Second row, (v) CD3+CD4+ cells are gated from R5 cells and then (vi) effectors (CD127$^{hi}$) and Tregs (CD127$^{low}$) CD25+ cells. (vii) gated Tregs from CD127$^{low}$FOXP3$^{hi}$ CD25+ cells. (viii–xii); same as (iii–vii) but from total CD45+ cells (i) and without exclusion of dead cells (ii).
(TIF)

**S2 Fig. Comparison of DNA quantities obtained using different lysis time during extraction procedure.** Results of non-parametric unpaired Mann-Whitney U tests are as shown $^{***} = p \leq 0.001$, $^{**} = p \leq 0.01$.
(TIF)

**S3 Fig. Comparison of flow cytometric and MS-qPCR data for CD3+ cell, Treg content, and Treg/CD3+ cell ratio in frozen samples with total cell gating by flow cytometry.** Cellular enumerations, from frozen CB segments, using flow cytometry were compared with epigenetic enumerations performed on the same samples. Male and female derived samples are as indicated. Flow cytometry assessments used total cell gating (gating without the exclusion of dead cells as shown in S1). **A;** CD3+ cells by flow cytometric or with TcSDR MS-qPCR enumeration. **B;** Treg by flow or with TSDR MS-qPCR enumeration. **C;** Ratio of Treg/CD3+ cells using the two methods.
(TIF)

**S4 Fig. CD3, Treg and Treg/CD3 ratio in fresh units and paired frozen segments.** Shown are MS-qPCR based enumerations in fresh samples from whole units (CBU), and paired samples from frozen segments (SE) from the same units. Shown is the results of Wilcoxon tests for paired observations.
(TIF)

**S1 Data. Flowjo data.**
(ZIP)

**S2 Data. Prism data.**
(ZIP)

## Acknowledgments

The authors would like to thank the Anthony Nolan Cell Therapy centre for their assistance.

## Author Contributions

**Conceptualization:** Richard C. Duggleby, Abigail A. Lamikanra, David J. Roberts, Diana Hernandez, J. Alejandro Madrigal, Robert D. Danby.

**Data curation:** Richard C. Duggleby.

**Formal analysis:** Richard C. Duggleby.

**Investigation:** Richard C. Duggleby, Hoi Pat Tsang, Kathryn Strange, Alasdair McWhinnie.

**Methodology:** Richard C. Duggleby.

**Supervision:** Diana Hernandez, J. Alejandro Madrigal, Robert D. Danby.

**Visualization:** Richard C. Duggleby.

**Writing – original draft:** Richard C. Duggleby, Diana Hernandez, Robert D. Danby.

**Writing – review & editing:** Richard C. Duggleby, Hoi Pat Tsang, Kathryn Strange, Alasdair McWhinnie, Abigail A. Lamikanra, David J. Roberts, Diana Hernandez, J. Alejandro Madrigal, Robert D. Danby.

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
