## [Decision Letter · Decision Letter 0]

24 Mar 2020

PONE-D-20-01315

Enumerating regulatory T cells in fresh and cryopreserved umbilical cord blood samples using FOXP3 methylation specific quantitative PCR.

PLOS ONE

Dear Dr Duggleby,

Thank you for submitting your manuscript to PLOS ONE. After careful consideration, we feel that it is potentially interesting but it does not fully meet PLOS ONE’s publication criteria in terms of priority as it currently stands. Therefore, we invite you to submit a revised version of the manuscript that addresses the ALL the points raised during the review process.

In particular, although 2 out of the 3 reviewers mention "minor modifications" their comments are relevant and some of them should be addressed by providing additional experimental data. The third reviewer recommended rejection based on general considerations concerning the rationale that you should also address in detail.

 Please consider that the revised version will be sent back to the original reviewers prior to any final decision is made.

May 08 2020 11:59PM. To enhance the reproducibility of your results, we recommend that if applicable you deposit your laboratory protocols in protocols.io, where a protocol can be assigned its own identifier (DOI) such that it can be cited independently in the future. For instructions see: http://journals.plos.org/plosone/s/submission-guidelines#loc-laboratory-protocols

We look forward to receiving your revised manuscript.

Yours Sincerely,

Lucienne Chatenoud

Academic Editor

PLOS ONE

Journal Requirements:

2. At this time, we ask that you please provide representative images of the flow cytometry dot plots for each Figure in which flow cytometry data is presented.

3. To comply with PLOS ONE submission guidelines, in your Methods section, please provide additional information regarding your statistical analyses. For more information on PLOS ONE's expectations for statistical reporting, please see https://journals.plos.org/plosone/s/submission-guidelines.#loc-statistical-reporting

4. Please confirm in your methods section and ethics statement that the 'East Midlands Derby Ethical Committee (15/EM/0045)' consists of a committee of experts that reviewed and approved your study.

Please include the full name of the IRB/ethics committee that reviewed and approved this study, including the name of the affiliated institution if applicable.

5. In the ethics statement in the manuscript and in the online submission form, please provide additional information about the patient records used in your retrospective study, including: a) whether all tissue samples were fully anonymized before you accessed them and b) the date range (month and year) during which patients' tissue samples were accessed.

6. Please provide additional details regarding participant consent. In the ethics statement in the Methods and online submission information, please ensure that you have specified what type of consent you obtained for the collection of CB units (for instance, written or verbal, and if verbal, how it was documented and witnessed).

Reviewers' comments:

Reviewer's Responses to Questions

**Comments to the Author**

1. Is the manuscript technically sound, and do the data support the conclusions?

Reviewer #1: Yes

Reviewer #2: Yes

Reviewer #3: Partly

2. Has the statistical analysis been performed appropriately and rigorously? 

Reviewer #1: Yes

Reviewer #2: Yes

Reviewer #3: Yes

3. Have the authors made all data underlying the findings in their manuscript fully available?

Reviewer #1: Yes

Reviewer #2: Yes

Reviewer #3: Yes

4. Is the manuscript presented in an intelligible fashion and written in standard English?

Reviewer #1: Yes

Reviewer #2: Yes

Reviewer #3: Yes

5. Review Comments to the Author

Reviewer #1: Duggleby et al. investigated the consistency and accuracy of flow cytometry and FOXP3/CD3 methylation status to assess the proportions of CD3+ and FOXP3+ Tregs in CB units and thawed segments. A previous study from the same group identified high Treg content in mobilized peripheral blood stem cell grafts as predictive of good post-transplant outcomes. The investigators aim to apply this regulatory T cell content assessment to transplantation from CBU. However, the low numbers and the poor viability of CB cells recovered from the stored segments (dedicated to quality check) raise significant challenges to tackle. To this end, they compared two approaches based either on FACS or epigenetic study. The main conclusions drawn from the study are the following:

1- There is a strong relationship between the flow cytometry and epigenetic assessment for the fresh CB cells. Molecular study requires less cells than FACS and as such may be a suitable alternative strategy.

2- With regard to thawed cells from frozen segments, the flow cytometry enumeration would become unreliable (due to the high cell death) and be outperformed by epigenetic assessment. The bottom line is that MS-qPCR captures epigenetic information even from dying cells, which are not excluded from the analysis.

However, one may think that capturing information about cell mortality following thawing could be relevant for graft assessment before patient administration. Indeed, I hardly see the advantage of counting the dying cells. In this respect, a missing information in the introduction might be worth to emphasize. The use of an attached segment to the CBU for quality check is still controversial. Indeed, previous studies showed that cell viabilities obtained from attached segments are not representative of those obtained from the bag. Cell mortality is usually far overrated in the attached segments, entailing illegitimate graft discard (Faivre L et al. Bone Marrow Transplantation 2017). Hence, a useful study would consist in correlating results obtained from FACS- and epigenetic-based strategies in the attached segments with those obtained from bag-derived thawed cells.

I have a few comments and suggestions:

1- Cord-blood-derived immuno-magnetically sorted CD4+ CD25+ T cells are presumed to be a pure population of Tregs and have been infused as such to HSCT patients. In contrast, CD4+ CD25+ T cells collected from peripheral blood are thought to be a mix of activated and regulatory T cells, and must be cultured with rapamycin before use, unless the purification is further refined (CD127, CD45RA). I am thus surprised to see a sizeable population of CD25+ CD127+ activated T cells in the CBU (Figure 1). Did the investigators included CD45RA / RO in the FACS panel?

2- As reminded in the discussion, cryopreservation negatively impacts on the ability to accurately gate Tregs with FACS, mostly because of CD25 downregulation in recently thawed cells. To overcome this issue, human PBMCs can be rested overnight at 37°C with IL-2 before cell staining (Savage et al. JCI Insight 2018).

3- Fig. 1A shows a dim FOXP3 expression in CD25+ CD127low cells. HELIOS staining, combined with FOXP3, might be useful to better discriminate genuine Tregs from activated T cells.

4- As discussed by authors, the greater proportion of CD3+ T cells, including Tregs, in frozen segments results from increased susceptibility to death of non CD3 cells after a freeze-thaw cycle. I wonder whether a few cell wash & spin cycles before epigenetic assessment would not get rid of dying cells and make results obtained from both techniques more comparable.

5- Page 19 : Why impaired XCI, leading to TSDR demethylation in the two FOXP3 loci, would result in autoimmune diseases?

Reviewer #2: The goal of the study is to evaluate epigenetic immune cell quantification in fresh and frozen umbilical cord blood (CB) samples to test the hypothesis that since quantification of CD3 and Treg in CB samples using flow cytometry is limited by available material and substantial cell death after freeze/thaw, the epigenetic enumeration might be a suitable alternative to flow cytometry.

Results show that:

Cell viability is strongly affected by freeze thawing of CB samples (Fig 1B)

Differences in enumeration between fresh and frozen samples are significantly higher when using flow cytometry than when using epigenetic enumeration (Fig. 2)

Epigenetic enumeration reproducibly yields higher levels of de-methylated TSDR in females vs. males (due to incomplete X-chromosome inactivation?) (Fig. 3)

In fresh samples, flow cytometry and epigenetic enumeration of CD3+ T cells correlate well for both males and females as well as for both sexes combined (Fig. 4A). Enumeration of Treg by both methods correlate in male and both sexes combined, but not in female samples (Fig. 4B). Treg/CD3 ratio only correlates in male samples but not in female and both sexes combined (Fig 4C).

In frozen samples, flow cytometry and epigenetic enumeration of CD3+ T-cells still show good correlation for males, females and both sexes combined. However, no correlation was seen for enumeration of Treg and Treg/CD3 ratio (Fig. 5)

Mann-Whitney is misspelled

Ref 12, in the text the fist author is misspelled

Human FOXP3 should be written capitalized

This work has absolutely no novelty and therefore limited impact but it is scientifically well done.

Reviewer #3: The authors proposed in their report a comparative study of two methods to assess Treg prevalence, i.e. flow cytometry and epigenetic study, in fresh and frozen cord blood. They concluded that epigenetics assessments are more practical and more accurate than flow cytometry.

Major comments:

1) Regarding the material and method, it is not clear whether both fresh and frozen samples were obtained from the same donors. Comparisons between frozen and corresponding fresh samples are mandatory for this kind of comparative study and paired t tests are required

2) Still I am puzzled by the assumption by the authors that it is difficult to do flow cytometry on fresh cord blood. Cord bloods are highly concentrated in t cells and even with very small volumes it is possible to do surface staining and analysis of T cell subsets.Therefore, I am not sure taht the rationale of such studies is justified by issues to address that actually do not exist

3) regarding flow cytometry analysis on thawed cells, mortality is matter of freezing methods quality. Anaylsis ofwhole DNA might overestimate the rate of live cells in the thawed cord blood while flow cytometry will enable the count of live cells, that are important at the clincal level.

Therefore I am unsure about the usefulness of the DNA methylation assays here

6. PLOS authors have the option to publish the peer review history of their article (what does this mean?). If published, this will include your full peer review and any attached files.

Reviewer #1: Yes: Dr Julien Zuber, MD, PhD

Reviewer #2: No

Reviewer #3: No

---

## [Author Response · Author response to Decision Letter 0]

16 Sep 2020

Editors comments

These will be amended during re-submission.

2. At this time, we ask that you please provide representative images of the flow cytometry dot plots for each Figure in which flow cytometry data is presented.

We have added an additional supplementary Figure (S1) to show a representative flow cytometry analysis of frozen CB segments, with and without exclusion of dead cells (S3), as we had only shown a representation of analysis of fresh CB.

3. To comply with PLOS ONE submission guidelines, in your Methods section, please provide additional information regarding your statistical analyses. For more information on PLOS ONE's expectations for statistical reporting, please see https://journals.plos.org/plosone/s/submission-guidelines.#loc-statistical-reporting

Line 321 (submission pdf); we have expanded on the description of the statistical methods to include the criteria for the statistical tests selected based on underlying assumptions.

4. Please confirm in your methods section and ethics statement that the 'East Midlands Derby Ethical Committee (15/EM/0045)' consists of a committee of experts that reviewed and approved your study.

Please include the full name of the IRB/ethics committee that reviewed and approved this study, including the name of the affiliated institution if applicable.

Paragraph line 201 (submission pdf); we have amended the paragraph describing the study population to indicate that the local ethics committee (of experts and lay persons) determined that non-clinical grade umbilical cord (CB) blood units and quality control segments can be used for research use only; we are not performing a study on patient samples, rather a study on non-clinical grade samples approved for research use.

5. In the ethics statement in the manuscript and in the online submission form, please provide additional information about the patient records used in your retrospective study, including: a) whether all tissue samples were fully anonymized before you accessed them and b) the date range (month and year) during which patients' tissue samples were accessed.

We did not use patient records but rather consented donors. All identifying data was anonymised. Donor data was not accessed other than to obtain the gender of the CB samples used. Paragraph starting line 201 has been altered to reflect this.

6. Please provide additional details regarding participant consent. In the ethics statement in the Methods and online submission information, please ensure that you have specified what type of consent you obtained for the collection of CB units (for instance, written or verbal, and if verbal, how it was documented and witnessed). 

Paragraph line 201 has been altered to indicate that written consent was obtained.

Reviewer #1

Duggleby et al. investigated the consistency and accuracy of flow cytometry and FOXP3/CD3 methylation status to assess the proportions of CD3+ and FOXP3+ Tregs in CB units and thawed segments. A previous study from the same group identified high Treg content in mobilized peripheral blood stem cell grafts as predictive of good post-transplant outcomes. The investigators aim to apply this regulatory T cell content assessment to transplantation from CBU. However, the low numbers and the poor viability of CB cells recovered from the stored segments (dedicated to quality check) raise significant challenges to tackle. To this end, they compared two approaches based either on FACS or epigenetic study. The main conclusions drawn from the study are the following:

1- There is a strong relationship between the flow cytometry and epigenetic assessment for the fresh CB cells. Molecular study requires less cells than FACS and as such may be a suitable alternative strategy.

2- With regard to thawed cells from frozen segments, the flow cytometry enumeration would become unreliable (due to the high cell death) and be outperformed by epigenetic assessment. The bottom line is that MS-qPCR captures epigenetic information even from dying cells, which are not excluded from the analysis.

However, one may think that capturing information about cell mortality following thawing could be relevant for graft assessment before patient administration. Indeed, I hardly see the advantage of counting the dying cells. In this respect, a missing information in the introduction might be worth to emphasize. The use of an attached segment to the CBU for quality check is still controversial. Indeed, previous studies showed that cell viabilities obtained from attached segments are not representative of those obtained from the bag. Cell mortality is usually far overrated in the attached segments, entailing illegitimate graft discard (Faivre L et al. Bone Marrow Transplantation 2017). 

Hence, a useful study would consist in correlating results obtained from FACS- and epigenetic-based strategies in the attached segments with those obtained from bag-derived thawed cells.

We thank the reviewer for their comments; however, we have clearly not conveyed the purpose of this study adequately. Consequently, we have made adjustments to the main text accordingly. The title, abstract, introduction and discussion have been reworked to emphasise the following points.

The ultimate future aim is to replicate the 2016 study [1] with peripheral blood mobilised stem cells (PBSC) in adult transplantation with single unit CB transplants. To perform this study using flow cytometry, samples would need to be taken directly from thawed CB units as they were transplanted, since, as the Faivre L et al. [2] indicates, the viability of segments does not accurately reflect the parent unit. However, this is not practical to obtain; prospectively this would be hard to get approval for due to the risk of compromising/contaminating the thawed CB unit being infused, especially at multiple sites. Secondly, a prospective study such as this would take a long time to acquire sufficient samples. Therefore, a retrospective study using historical CB samples and clinical data is the only realistic alternative, which means the only source of material is isolated DNA or cryopreserved CB segments. This would also be the only source of material if this parameter became one of the selection criteria for CB units.

 Thus:-

• The first aim of this study was to validate the DNA based methodology. Whilst this has been performed in adult and, in part, in CB it has not been used to measure the Treg/CD3 cell ratio in cryopreserved CB segments. To this end. Tregs and CD3 cells were assessed in the optimal conditions of freshly collected CB units where flow cytometry and DNA enumerations should be comparable. 

• The second aim was to determine if DNA based measurement of the Treg/CD3 ratio was a robust measurement in cryopreserved segments. 

Additionally, addressing the point about capturing the mortality in the unit, we are ultimately seeking only to measuring the Treg/CD3 ratio. This is important as we are not trying the measure absolute numbers (which would require viability to be accounted for). Instead the ratio between Tregs and CD3 is considered across different samples. It is this measure that will ultimately be compared against clinical outcome, splitting the units into “high” and “low” risk groups in a similar manner to the earlier Danby et al study in PBSC transplants[1]. 

To address the above points the abstract, has been modified. 

Line 86 (submission pdf); the introduction has been modified to address Tregs in HCT including the problems enumerating Tregs in CB banking. Paragraph starting line 136 has been moved down and modified for clarity. Paragraph starting line 159 has been modified to make the aims of the study clear.

The results section has been re-worked to emphasise these points in (line 334-407, 425-453, 499-538).

The Discussion starting line 557 has been also re-worked to emphasise these points.

I have a few comments and suggestions:

1- Cord-blood-derived immuno-magnetically sorted CD4+ CD25+ T cells are presumed to be a pure population of Tregs and have been infused as such to HSCT patients. In contrast, CD4+ CD25+ T cells collected from peripheral blood are thought to be a mix of activated and regulatory T cells, and must be cultured with rapamycin before use, unless the purification is further refined (CD127, CD45RA). I am thus surprised to see a sizeable population of CD25+ CD127+ activated T cells in the CBU (Figure 1). Did the investigators included CD45RA / RO in the FACS panel?

To directly address the first point, in the example shown in the Figure 1A 2.75% of the cells are shown to be CD25+CD127+ compared with 6.59% CD25+CD127low (shown to have higher FOXP3+ than CD25+CD127+ effectors). By contrast, adult peripheral blood contains around 40% CD25+CD127+ cells (Duggleby et al. Methods in Molecular Biology 2014 [3]). In this example, we observed similar numbers (2.1%) of CD127+CD25+ cells in the CB samples. Seddiki et al. 2006 [4] also shows a similar, if not higher proportion of CD127hiCD25+ cells in their characterisation of the CB. They show that this population has intermediate FOXP3 levels and is not suppressive.

With regards to the second point about using CD45RA. Previous studies have observed, like us, that there are very low numbers of CD45RA- cells in the CB (e.g. Figueroa-Tentori et al 2008 [5]). Fujimaki et al. (2008) [5] shows that the CD45RO+ population is both small and much lower intensity of expression compared with peripheral blood. We have, in the past characterised this CD127hiCD25+ population for CD45RA expression (unpublished observations). We found that it was no better at distinguishing the Treg population than combining the CD127- and FOXP3+ expression. 

2- As reminded in the discussion, cryopreservation negatively impacts on the ability to accurately gate Tregs with FACS, mostly because of CD25 downregulation in recently thawed cells. To overcome this issue, human PBMCs can be rested overnight at 37°C with IL-2 before cell staining (Savage et al. JCI Insight 2018).

Thank you for the suggestion, however, since the segments contain a very high proportion dead and dying cells, this is likely to cause even more cell death. Unfortunately, the cryopreserved CB units are not a mononuclear cell preparation, they are volume reduced sample. Consequently, they contain an intact granulocyte population (mostly dead and dying post thaw). Additionally, any further manipulation is likely to result in further loss of very limited material. 

3- Fig. 1A shows a dim FOXP3 expression in CD25+ CD127low cells. HELIOS staining, combined with FOXP3, might be useful to better discriminate genuine Tregs from activated T cells.

Again, thank you for the input. We have found that Helios can also be present on activated T cells. We are in the process of publishing helios expression on expanded Tregs and effector cells (in preparation). Effector cells show intermediate Helios expression and Tregs helioshi expression. Thus, the use of these markers is not quite as clear cut as we would like. Additionally, as with CD45RA, the main issue when assessing segments was the lack of material.

4- As discussed by authors, the greater proportion of CD3+ T cells, including Tregs, in frozen segments results from increased susceptibility to death of non CD3 cells after a freeze-thaw cycle. I wonder whether a few cell wash & spin cycles before epigenetic assessment would not get rid of dying cells and make results obtained from both techniques more comparable.

Thank you for the suggestion, however, it has been our experience that only methods such as dead cell removal kits from Miltenyi Biotec can approach efficacy with frozen CB, but not without significant loss of material.

5- Page 19 : Why impaired XCI, leading to TSDR demethylation in the two FOXP3 loci, would result in autoimmune diseases?

We have modified the discussion (line 610-632); this point was removed to make the discussion more concise. However, to answer the query, the reasoning is that under non-pathological conditions, female subjects inactivate the FOXP3 copy that is mutated. IPEX females, for example, are heterozygous for the mutated FOXP3 (on the X chromosome) and inactivate the faulty copy [6]. Hence only males suffer from IPEX syndrome. This means that the XCI is under some sort of positive feedback and can act as a quality control for the immune genes (in fact many of the immune genes are X-chromosome linked). As females get older more errors can occur in the FOXP3 gene. Normally these would be inactivated by XCI (assuming both FOXP3 copies are not affected). However, reduced XCI function is associated with age. Consequently, Tregs can be generated with impaired functioning FOXP3 [7]. Since the size of the Treg niche is fixed this means that the average function of the Treg pool is lower. This may, therefore, explain why age and gender are contributing factors to the aetiology of many autoimmune diseases (such as Rheumatoid arthritis) [8,9]. 

Reviewer #2: The goal of the study is to evaluate epigenetic immune cell quantification in fresh and frozen umbilical cord blood (CB) samples to test the hypothesis that since quantification of CD3 and Treg in CB samples using flow cytometry is limited by available material and substantial cell death after freeze/thaw, the epigenetic enumeration might be a suitable alternative to flow cytometry.

Results show that:

Cell viability is strongly affected by freeze thawing of CB samples (Fig 1B)

Differences in enumeration between fresh and frozen samples are significantly higher when using flow cytometry than when using epigenetic enumeration (Fig. 2)

Epigenetic enumeration reproducibly yields higher levels of de-methylated TSDR in females vs. males (due to incomplete X-chromosome inactivation?) (Fig. 3)

In fresh samples, flow cytometry and epigenetic enumeration of CD3+ T cells correlate well for both males and females as well as for both sexes combined (Fig. 4A). Enumeration of Treg by both methods correlate in male and both sexes combined, but not in female samples (Fig. 4B). Treg/CD3 ratio only correlates in male samples but not in female and both sexes combined (Fig 4C).

In frozen samples, flow cytometry and epigenetic enumeration of CD3+ T-cells still show good correlation for males, females and both sexes combined. However, no correlation was seen for enumeration of Treg and Treg/CD3 ratio (Fig. 5)

Mann-Whitney is misspelled

Noted, this has been corrected in the main text.

Ref 12, in the text the first author is misspelled

Thank you, corrected from Lui to Liu.

Human FOXP3 should be written capitalized

Noted, this has been corrected in the main text.

This work has absolutely no novelty and therefore limited impact but it is scientifically well done. 

Thank you for your input. Whilst this type of assessment has been performed in adults and, in part, in CB, it has not been used in cryopreserved CB segments. Additionally, it has not been used to measure the Treg/CD3 in the CB segments. If DNA can be used in historical samples this allows for retrospective analysis on clinical samples and subsequently increased chance of achieving a meaningful sample size. Ultimately, if a relationship between Treg/CD3 ratio and graft outcome is demonstrated in CB transplantation, as it has with adult PBSC transplantation, then this may become a novel selection criterion during CB banking. 

reviewer #3: The authors proposed in their report a comparative study of two methods to assess Treg prevalence, i.e. flow cytometry and epigenetic study, in fresh and frozen cord blood. They concluded that epigenetics assessments are more practical and more accurate than flow cytometry.

Major comments:

1) Regarding the material and method, it is not clear whether both fresh and frozen samples were obtained from the same donors. Comparisons between frozen and corresponding fresh samples are mandatory for this kind of comparative study and paired t tests are required

The samples were not paired, a large number of fresh samples were compared with frozen segment samples. Whilst ideally, we would want paired data, for these observations it would require the CB segments to be inherently different from the units. However, we do have epigenetic data from segments paired to the parent unit showing the same result as unpaired data, indicating that the samples were not inherently different. This data has been added as an additional supplementary figure (S4) and included in the text of the discussion (line 711). 

2) Still I am puzzled by the assumption by the authors that it is difficult to do flow cytometry on fresh cord blood. Cord bloods are highly concentrated in t cells and even with very small volumes it is possible to do surface staining and analysis of T cell subsets. Therefore, I am not sure that the rationale of such studies is justified by issues to address that actually do not exist.

Thank you for your comment, we have obviously not been clear that we did indeed have no issue with assessing the Treg/CD3 ratio in the fresh samples. the main text has been altered to emphasise that one of the aims of this study was to validate the use of the epigenetic base method in fresh CB samples, where the assessment by flow cytometry is optimal and thus a high degree of agreement expected between the two measures. The discussion has also been modified to emphasise that whilst these two measurements did not completely agree (we believe due to inherent differences in the methodologies) there is a strong degree of correlation, indicating that they are related measures (line 567-587).

3) regarding flow cytometry analysis on thawed cells, mortality is matter of freezing methods quality. Analysis of whole DNA might overestimate the rate of live cells in the thawed cord blood while flow cytometry will enable the count of live cells, that are important at the clinical level.

Therefore I am unsure about the usefulness of the DNA methylation assays here.

Thank you for your comment and this is similar to that raised by reviewer #1. Thus, we have modified the introduction to emphasise that our aim is not the absolute enumeration of Tregs and CD3 cells but rather the relative proportions of them in a transplanted unit. Since we are measuring the ratio rather than absolute numbers, we can overcome the inherent variance in thawed samples. It is this measure that will ultimately be compared against clinical outcome, splitting the units into “high” and “low” risk groups in a similar manner to the earlier Danby et al study in PBSC transplants. The hypothesis is that the influence of Tregs and T cells on transplant outcome is relatively strong. If true, this may provide a method that can be used as a selection criterion for CB units, using a sample outside of the main body of the unit in storage. In CB banking these are the only means of assessing a CB unit in storage.

References

1. Danby RD, Zhang W, Medd P, Littlewood TJ, Peniket A, Rocha V, et al. High proportions of regulatory T cells in PBSC grafts predict improved survival after allogeneic haematopoietic SCT. Bone Marrow Transplant. 2016;51(1):110–8. doi: 10.1038/bmt.2015.215

2. Faivre L, Boucher H, Zerbib R, Domet T, Desproges A, Couzin C, et al. Cord blood attached segment: is this a relevant quality control to predict a good hematopoietic stem cell graft? Bone Marrow Transplant. 2017;52(9):1353–4. doi: 10.1038/bmt.2017.150

3. Duggleby RC, Madrigal JAA. Methods of detection of immune reconstitution and T regulatory cells by flow cytometry. In: Beksaç M, editor. Bone Marrow and Stem Cell Transplantation. New York, NY: Springer New York; 2014. p. 159–86. doi: 10.1007/978-1-4614-9437-9_10

4. Seddiki N, Santner-Nanan B, Martinson J, Zaunders J, Sasson S, Landay A, et al. Expression of interleukin (IL)-2 and IL-7 receptors discriminates between human regulatory and activated T cells. J Exp Med. 2006 Jul;203(7):1693–700. doi: 10.1084/jem.20060468

5. Figueroa-Tentori D, Querol S, Dodi IA, Madrigal A, Duggleby R. High purity and yield of natural Tregs from cord blood using a single step selection method. J Immunol Methods. 2008 Dec 31;339(2):228–35. doi: 10.1016/j.jim.2008.09.019

6. Tommasini A, Ferrari S, Moratto D, Badolato R, Boniotto M, Pirulli D, et al. X-chromosome inactivation analysis in a female carrier of FOXP3 mutation. Clin Exp Immunol. 2002 Oct;130(1):127–30. doi: 10.1046/j.1365-2249.2002.01940.x

7. Broen JCA, Wolvers-Tettero ILM, Geurts-Van Bon L, Vonk MC, Coenen MJH, Lafyatis R, et al. Skewed X chromosomal inactivation impacts T regulatory cell function in systemic sclerosis. Ann Rheum Dis. 2010;69(12):2213–6. doi: 10.1136/ard.2010.129999

8. Wang J, Syrett CM, Kramer MC, Basu A, Atchison ML, Anguera MC. Unusual maintenance of X chromosome inactivation predisposes female lymphocytes for increased expression from the inactive X. Proc Natl Acad Sci. 2016;113(14):E2029–38. doi: 10.1073/pnas.1520113113

9. Syrett CM, Sindhava V, Sierra I, Dubin AH, Atchison M, Anguera MC. Diversity of Epigenetic Features of the Inactive X-Chromosome in NK Cells, Dendritic Cells, and Macrophages. Front Immunol. 2018;9(January):3087. doi: 10.3389/fimmu.2018.03087

---

## [Editor Report · Decision Letter 1]

22 Sep 2020

Enumerating regulatory T cells in cryopreserved umbilical cord blood samples using FOXP3 methylation specific quantitative PCR

PONE-D-20-01315R1

Dear Dr. Duggleby,

We are pleased to inform you that the revised version of your manuscript has been judged scientifically suitable for publication and will be formally accepted for publication once it meets all outstanding technical requirements.

Yours Sincerely,

Lucienne Chatenoud

Academic Editor

PLOS ONE
---

## [Editor Report · Acceptance letter]

12 Oct 2020

PONE-D-20-01315R1 

Enumerating regulatory T cells in cryopreserved umbilical cord blood samples using *FOXP3* methylation specific quantitative PCR 

Dear Dr. Duggleby:

I'm pleased to inform you that your manuscript has been deemed suitable for publication in PLOS ONE. Congratulations! Your manuscript is now with our production department. 

Kind regards, 

on behalf of

Professor Lucienne Chatenoud 

Academic Editor

PLOS ONE